# Adaptive Accelerated Gradient Converging Method under Hölderian Error Bound Condition

**Mingrui Liu, Tianbao Yang**
Department of Computer Science
The University of Iowa, Iowa City, IA 52242
`mingrui-liu, tianbao-yang@uiowa.edu`

## Abstract

Recent studies have shown that proximal gradient (PG) method and accelerated gradient method (APG) with restarting can enjoy a linear convergence under a weaker condition than strong convexity, namely a quadratic growth condition (QGC). However, the faster convergence of restarting APG method relies on the potentially unknown constant in QGC to appropriately restart APG, which restricts its applicability. We address this issue by developing a novel adaptive gradient converging methods, i.e., leveraging the magnitude of proximal gradient as a criterion for restart and termination. Our analysis extends to a much more general condition beyond the QGC, namely the Hölderian error bound (HEB) condition. *The key technique* for our development is a novel synthesis of *adaptive regularization and a conditional restarting scheme*, which extends previous work focusing on strongly convex problems to a much broader family of problems. Furthermore, we demonstrate that our results have important implication and applications in machine learning: (i) if the objective function is coercive and semi-algebraic, PG's convergence speed is essentially $o(\frac{1}{t})$, where $t$ is the total number of iterations; (ii) if the objective function consists of an $\ell_1$, $\ell_\infty$, $\ell_{1,\infty}$, or huber norm regularization and a convex smooth piecewise quadratic loss (e.g., square loss, squared hinge loss and huber loss), the proposed algorithm is parameter-free and enjoys a *faster linear convergence* than PG without any other assumptions (e.g., restricted eigen-value condition). It is notable that our linear convergence results for the aforementioned problems are global instead of local. To the best of our knowledge, these improved results are first shown in this work.

## 1 Introduction

We consider the following smooth composite optimization:

$$\min_{\mathbf{x} \in \mathbb{R}^d} F(\mathbf{x}) \triangleq f(\mathbf{x}) + g(\mathbf{x}), \tag{1}$$

where $g(\mathbf{x})$ is a proper lower semi-continuous convex function and $f(\mathbf{x})$ is a continuously differentiable convex function, whose gradient is $L$-Lipschitz continuous. The above problem has been studied extensively in literature and many algorithms have been developed with convergence guarantee. In particular, by employing the proximal mapping associated with $g(\mathbf{x})$, i.e.,

$$P_{\eta g}(\mathbf{u}) = \arg \min_{\mathbf{x} \in \mathbb{R}^d} \frac{1}{2}\|\mathbf{x} - \mathbf{u}\|_2^2 + \eta g(\mathbf{x}), \tag{2}$$

proximal gradient (PG) and accelerated proximal gradient (APG) methods have been developed for solving (1) with $O(1/\epsilon)$ and $O(1/\sqrt{\epsilon})$ [1] iteration complexities for finding an $\epsilon$-optimal solution.

Table 1: Summary of iteration complexities in this work under the HEB condition with $\theta \in (0, 1/2]$, where $G(\mathbf{x})$ denotes the proximal gradient, $\mathcal{C}(1/\epsilon^\alpha) = \max(1/\epsilon^\alpha, \log(1/\epsilon))$ and $\widetilde{O}(\cdot)$ suppresses a logarithmic term. If $\theta > 1/2$, all algorithms can converge with finite steps of proximal mapping. rAPG stands for restarting APG. $*$ mark results available for certain subclasses of problems.

| algo. | PG | rAPG | adaAGC |
|---|---|---|---|
| $F(\mathbf{x}) - F_* \leq \epsilon$ | $O\left(c^2 L\mathcal{C}\left(\frac{1}{\epsilon^{1-2\theta}}\right)\right)$ | $O\left(c\sqrt{L}\mathcal{C}\left(\frac{1}{\epsilon^{1/2-\theta}}\right)\right)$ | $*$ |
| $\|G(\mathbf{x})\|_2 \leq \epsilon$ | $O\left(c^{\frac{1}{1-\theta}} L\mathcal{C}\left(\frac{1}{\epsilon^{\frac{1-2\theta}{1-\theta}}}\right)\right)$ | – | $\widetilde{O}\left(c^{\frac{1}{2(1-\theta)}}\sqrt{L}\mathcal{C}\left(\frac{1}{\epsilon^{\frac{1-2\theta}{2(1-\theta)}}}\right)\right)$ |
| requires $\theta$ | No | Yes | Yes |
| requires $c$ | No | Yes | No |

When either $f(\mathbf{x})$ or $g(\mathbf{x})$ is strongly convex, both PG and APG can enjoy a linear convergence, i.e., the iteration complexity is improved to be $O(\log(1/\epsilon))$.

Recently, a wave of studies try to generalize the linear convergence to problems without strong convexity but under certain structured condition of the objective function or more generally a quadratic growth condition [8, 32, 21, 23, 7, 31, 3, 15, 9, 29, 4, 24, 26, 25]. Earlier work along the line dates back to [12, 13, 14]. An example of the structured condition is such that $f(\mathbf{x}) = h(A\mathbf{x})$ where $h(\cdot)$ is strongly convex function and $\nabla h(\mathbf{x})$ is Lipschitz continuous on any compact set, and $g(\mathbf{x})$ is a polyhedral function. Under such a structured condition, a local error bound condition can be established [12, 13, 14], which renders an asymptotic (local) linear convergence for the proximal gradient method. A quadratic growth condition (QGC) prescribes that the objective function satisfies for any $\mathbf{x} \in \mathbb{R}^d$ [2]: $\frac{\alpha}{2}\|\mathbf{x} - \mathbf{x}_*\|_2^2 \leq F(\mathbf{x}) - F(\mathbf{x}_*)$, where $\mathbf{x}_*$ denotes a closest point to $\mathbf{x}$ in the optimal set. Under such a quadratic growth condition, several recent studies have established the linear convergence of PG, APG and many other algorithms (e.g., coordinate descent methods) [3, 15, 4, 9, 29]. A notable result is that PG enjoys an iteration complexity of $O(\frac{L}{\alpha}\log(1/\epsilon))$ without knowing the value of $\alpha$, while a restarting version of APG studied in [15] enjoys an improved iteration complexity of $O(\sqrt{\frac{L}{\alpha}}\log(1/\epsilon))$ hinging on the value of $\alpha$ to appropriately restart APG periodically. Other equivalent conditions or more restricted conditions are also considered in several studies to show the linear convergence of (proximal) gradient method and other methods [9, 15, 29, 30].

In this paper, we extend this line of work to a more general error bound condition, i.e., the Hölderian error bound (HEB) condition on a compact sublevel set $\mathcal{S}_\xi = \{\mathbf{x} \in \mathbb{R}^d : F(\mathbf{x}) - F(\mathbf{x}_*) \leq \xi\}$: there exists $\theta \in (0, 1]$ and $0 < c < \infty$ such that

$$\|\mathbf{x} - \mathbf{x}_*\|_2 \leq c(F(\mathbf{x}) - F(\mathbf{x}_*))^\theta, \ \forall \mathbf{x} \in \mathcal{S}_\xi. \tag{3}$$

Note that when $\theta = 1/2$ and $c = \sqrt{1/\alpha}$, the HEB reduces to the QGC. In the sequel, we will refer to $C = Lc^2$ as condition number of the problem. It is worth mentioning that Bolte et al. [3] considered the same condition or an equivalent Kurdyka - Łojasiewicz inequality but they only focused on descent methods that bear a sufficient decrease condition for each update consequentially excluding APG. In addition, they do not provide explicit iteration complexity under the general HEB condition.

As a warm-up and motivation, we will first present a straightforward analysis to show that PG is automatically adaptive and APG can be made adaptive to the HEB by restarting. In particular if $F(\mathbf{x})$ satisfies a HEB condition on the initial sublevel set, PG has an iteration complexity of $O(\max(\frac{C}{\epsilon^{1-2\theta}}, C\log(\frac{1}{\epsilon})))$ [3], and restarting APG enjoys an iteration complexity of $O(\max(\frac{\sqrt{C}}{\epsilon^{1/2-\theta}}, \sqrt{C}\log(\frac{1}{\epsilon})))$ for the convergence of objective value, where $C = Lc^2$ is the condition number. These two results resemble but generalize recent works that establish linear convergence of PG and restarting APG under the QGC - a special case of HEB. Although enjoying faster convergence, restarting APG has a critical caveat: it requires the knowledge of constant $c$ in HEB to restart APG, which is usually difficult to compute or estimate. In this paper, we make nontrivial contributions to

obtain faster convergence of the proximal gradient's norm under the HEB condition by developing an adaptive accelerated gradient converging method.

The main results of this paper are summarized in Table 1. The contributions of this paper are: (i) we extend the analysis of PG and restarting APG under the quadratic growth condition to more general HEB condition, and establish the adaptive iteration complexities of both algorithms; (ii) to enjoy faster convergence of restarting APG and to eliminate the algorithmic dependence on the unknown parameter $c$, we propose and analyze an adaptive accelerated gradient converging (adaGC) method. The developed algorithms and theory have important implication and applications in machine learning. Firstly, if the considered objective function is also coercive and semi-algebraic (e.g., a norm regularized problem in machine learning with a semi-algebraic loss function), then PG's convergence speed is essentially $o(1/t)$ instead of $O(1/t)$, where $t$ is the total number of iterations. Secondly, for solving $\ell_1$, $\ell_\infty$ or $\ell_{1,\infty}$ regularized smooth loss minimization problems including least-squares loss, squared hinge loss and huber loss, the proposed adaGC method enjoys a linear convergence and a square root dependence on the "condition" number. In contrast to previous work, the proposed algorithm is parameter free and does not rely on any restricted conditions (e.g., the restricted eigen-value conditions).

## 2 Notations and Preliminaries

In this section, we present some notations and preliminaries. In the sequel, we let $\|\cdot\|_p$ ($p \geq 1$) denote the $p$-norm of a vector. A function $g(\mathbf{x}) : \mathbb{R}^d \to (-\infty, \infty]$ is a proper function if $g(\mathbf{x}) < +\infty$ for at least one $\mathbf{x}$. $g(\mathbf{x})$ is lower semi-continuous at a point $\mathbf{x}_0$ if $\liminf_{\mathbf{x} \to \mathbf{x}_0} g(\mathbf{x}) = g(\mathbf{x}_0)$. A function $F(\mathbf{x})$ is coercive if and only if $F(\mathbf{x}) \to \infty$ as $\|\mathbf{x}\|_2 \to \infty$. We will also refer to semi-algebraic set and semi-algebraic function several times in the paper, which are standard concepts in mathematics [2]. Due to limit of space, we present the definitions in the supplement.

Denote by $\mathbb{N}$ the set of all positive integers. A function $h(\mathbf{x})$ is a real polynomial if there exists $r \in \mathbb{N}$ such that $h(\mathbf{x}) = \sum_{0 \leq |\alpha| \leq r} \lambda_\alpha \mathbf{x}^\alpha$, where $\lambda_\alpha \in \mathbb{R}$ and $\mathbf{x}^\alpha = x_1^{\alpha_1} \ldots x_d^{\alpha_d}$, $\alpha_j \in \mathbb{N} \cup \{0\}$, $|\alpha| = \sum_{j=1}^d \alpha_j$ and $r$ is referred to as the degree of $h(\mathbf{x})$. A continuous function $f(\mathbf{x})$ is said to be a piecewise convex polynomial if there exist finitely many polyhedra $P_1, \ldots, P_k$ with $\cup_{j=1}^k P_j = \mathbb{R}^n$ such that the restriction of $f$ on each $P_j$ is a convex polynomial. Let $f_j$ be the restriction of $f$ on $P_j$. The degree of a piecewise convex polynomial function $f$ denoted by $deg(f)$ is the maximum of the degree of each $f_j$. If $deg(f) = 2$, the function is referred to as a piecewise convex quadratic function. Note that a piecewise convex polynomial function is not necessarily a convex function [10].

A function $f(\mathbf{x})$ is $L$-smooth w.r.t $\|\cdot\|_2$ if it is differentiable and has a Lipschitz continuous gradient with the Lipschitz constant $L$, i.e., $\|\nabla f(\mathbf{x}) - \nabla f(\mathbf{y})\|_2 \leq L\|\mathbf{x} - \mathbf{y}\|_2, \forall \mathbf{x}, \mathbf{y}$. Let $\partial g(\mathbf{x})$ denote the subdifferential of $g$ at $\mathbf{x}$. Denote by $\|\partial g(\mathbf{x})\|_2 = \min_{\mathbf{u} \in \partial g(\mathbf{x})} \|\mathbf{u}\|_2$. A function $g(\mathbf{x})$ is $\alpha$-strongly convex w.r.t $\|\cdot\|_2$ if it satisfies for any $\mathbf{u} \in \partial g(\mathbf{y})$ such that $g(\mathbf{x}) \geq g(\mathbf{y}) + \mathbf{u}^\top(\mathbf{x} - \mathbf{y}) + \frac{\alpha}{2}\|\mathbf{x} - \mathbf{y}\|_2^2, \forall \mathbf{x}, \mathbf{y}$.

Denote by $\eta > 0$ a positive scalar, and let $P_{\eta g}$ be the proximal mapping associated with $\eta g(\cdot)$ defined in (2). Given an objective function $F(\mathbf{x}) = f(\mathbf{x}) + g(\mathbf{x})$, where $f(\mathbf{x})$ is $L$-smooth and convex, $g(\mathbf{x})$ is a simple non-smooth function which is closed and convex, define a **proximal gradient** $G_\eta(\mathbf{x})$ as:

$$G_\eta(\mathbf{x}) = \frac{1}{\eta}(\mathbf{x} - \mathbf{x}_\eta^+), \text{ where } \mathbf{x}_\eta^+ = P_{\eta g}(\mathbf{x} - \eta \nabla f(\mathbf{x})).$$

When $g(\mathbf{x}) = 0$, we have $G_\eta(\mathbf{x}) = \nabla f(\mathbf{x})$, i.e., the proximal gradient is the gradient. It is known that $\mathbf{x}$ is an optimal solution iff $G_\eta(\mathbf{x}) = 0$. If $\eta = 1/L$, for simplicity we denote by $G(\mathbf{x}) = G_{1/L}(\mathbf{x})$ and $\mathbf{x}^+ = P_{g/L}(\mathbf{x} - \nabla f(\mathbf{x})/L)$. Let $F_*$ denote the optimal objective value to $\min_{\mathbf{x} \in \mathbb{R}^d} F(\mathbf{x})$ and $\Omega_*$ denote the optimal set. Denote by $\mathcal{S}_\xi = \{\mathbf{x} : F(\mathbf{x}) - F_* \leq \xi\}$ the $\xi$-sublevel set of $F(\mathbf{x})$. Let $D(\mathbf{x}, \Omega) = \min_{\mathbf{y} \in \Omega} \|\mathbf{x} - \mathbf{y}\|_2$.

The proximal gradient (PG) method solves the problem (1) by the update

$$\mathbf{x}_{t+1} = P_{\eta g}(\mathbf{x}_t - \eta \nabla f(\mathbf{x}_t)), \tag{4}$$

with $\eta \leq 1/L$ starting from some initial solution $\mathbf{x}_1 \in \mathbb{R}^d$. It can be shown that PG has an iteration complexity of $O(\frac{LD(\mathbf{x}_1, \Omega_*)^2}{\epsilon})$. Nevertheless, accelerated proximal gradient (APG) converges faster than PG. There are many variants of APG in literature [22] including the well-known FISTA [1]. The

**Algorithm 1:** ADG
---
$\mathbf{x}_0 \in \Omega,\ A_0 = 0,\ \mathbf{v}_0 = \mathbf{x}_0$
**for** $t = 0, \ldots, T$ **do**

   | Find $a_{t+1}$ from quadratic equation $\frac{a^2}{A_t + a} = 2\frac{1 + \alpha A_t}{L}$
   | Set $A_{t+1} = A_t + a_{t+1}$
   | Set $\mathbf{y}_t = \frac{A_t}{A_{t+1}}\mathbf{x}_t + \frac{a_{t+1}}{A_{t+1}}\mathbf{v}_t$
   | Compute $\mathbf{x}_{t+1} = P_{g/L}(\mathbf{y}_t - \nabla f(\mathbf{y}_t)/L)$
   | Compute $\mathbf{v}_{t+1} = \arg\min_{\mathbf{x}} \sum_{\tau=1}^{t+1} a_\tau \nabla f(\mathbf{x}_\tau)^\top \mathbf{x} + A_{t+1}g(\mathbf{x}) + \frac{1}{2}\|\mathbf{x} - \mathbf{x}_0\|_2^2$
---

simplest variant adopts the following update

$$\mathbf{y}_t = \mathbf{x}_t + \beta_t(\mathbf{x}_t - \mathbf{x}_{t-1}),\ \mathbf{x}_{t+1} = P_{\eta g}(\mathbf{y}_t - \eta \nabla f(\mathbf{y}_t)),$$

where $\eta \leq 1/L$ and $\beta_t$ is an appropriate sequence (e.g. $\beta_t = \frac{t-1}{t+2}$). APG enjoys an iteration complexity of $O(\frac{\sqrt{L}D(\mathbf{x}_1, \Omega_*)}{\sqrt{\epsilon}})$ [22]. Furthermore, if $f(\mathbf{x})$ is both $L$-smooth and $\alpha$-strongly convex, one can set $\beta_t = \frac{\sqrt{L} - \sqrt{\alpha}}{\sqrt{L} + \sqrt{\alpha}}$ and deduce a linear convergence [16, 11] with a better dependence on the condition number than that of PG. If $g(\mathbf{x})$ is $\alpha$-strongly convex and $f(\mathbf{x})$ is $L$-smooth, Nesterov [17] proposed a different variant based on dual averaging, which is referred to accelerated dual gradient (ADG) method and will be useful for our development. The key steps are presented in Algorithm 1.

## 2.1 Hölderian error bound (HEB) condition

**Definition 1** (Hölderian error bound (HEB)). *A function $F(\mathbf{x})$ is said to satisfy a HEB condition on the $\xi$-sublevel set if there exist $\theta \in (0, 1]$ and $0 < c < \infty$ such that for any $\mathbf{x} \in \mathcal{S}_\xi$*

$$dist(\mathbf{x}, \Omega_*) \leq c(F(\mathbf{x}) - F_*)^\theta. \tag{5}$$

The HEB condition is closely related to the Łojasiewicz inequality or more generally Kurdyka-Łojasiewicz (KL) inequality in real algebraic geometry. It has been shown that when functions are semi-algebraic and continuous, the above inequality is known to hold on any compact set [3]. We refer the readers to [3] for more discussions on HEB and KL inequalities.

In the remainder of this section, we will review some previous results to demonstrate that HEB is a generic condition that holds for a broad family of problems of interest. The following proposition states that any proper, coercive, convex, lower-semicontinuous and semi-algebraic functions satisfy the HEB condition.

**Proposition 1.** *[3] Let $F(\mathbf{x})$ be a proper, coercive, convex, lower semicontinuous and semi-algebraic function. Then there exists $\theta \in (0, 1]$ and $0 < c < \infty$ such that $F(\mathbf{x})$ satisfies the HEB on any $\xi$-sublevel set.*

**Example:** Most optimization problems in machine learning with an objective that consists of an empirical loss that is semi-algebraic (e.g., hinge loss, squared hinge loss, absolute loss, square loss) and a norm regularization $\|\cdot\|_p$ ($p \geq 1$ is a rational) or a norm constraint are proper, coercive, lower semicontinuous and semi-algebraic functions.

Next two propositions exhibit the value $\theta$ for piecewise convex quadratic functions and piecewise convex polynomial functions.

**Proposition 2.** *[10] Let $F(\mathbf{x})$ be a piecewise convex quadratic function on $\mathbb{R}^d$. Suppose $F(\mathbf{x})$ is convex. Then for any $\xi > 0$, there exists $0 < c < \infty$ such that $D(\mathbf{x}, \Omega_*) \leq c(F(\mathbf{x}) - F_*)^{1/2}, \forall \mathbf{x} \in \mathcal{S}_\xi$.*

Many problems in machine learning are piecewise convex quadratic functions, which will be discussed more in Section 5.

**Proposition 3.** *[10] Let $F(\mathbf{x})$ be a piecewise convex polynomial function on $\mathbb{R}^d$. Suppose $F(\mathbf{x})$ is convex. Then for any $\xi > 0$, there exists $c > 0$ such that $D(\mathbf{x}, \Omega_*) \leq c(F(\mathbf{x}) - F_*)^{\frac{1}{(deg(F)-1)^d+1}}, \forall \mathbf{x} \in \mathcal{S}_\xi$.*

---

**Algorithm 2:** restarting APG (rAPG)

---

**Input**: the number of stages $K$ and $\mathbf{x}_0 \in \Omega$
**for** $k = 1, \dots, K$ **do**
    Set $\mathbf{y}_1^k = \mathbf{x}_{k-1}$ and $\mathbf{x}_1^k = \mathbf{x}_{k-1}$
    **for** $\tau = 1, \dots, t_k$ **do**
        Update $\mathbf{x}_{\tau+1}^k = P_{g/L}(\mathbf{y}_\tau^k - \nabla f(\mathbf{y}_\tau^k)/L)$
        Update $\mathbf{y}_{\tau+1}^k = \mathbf{x}_{\tau+1}^k + \frac{\tau}{\tau+3}(\mathbf{x}_{\tau+1}^k - \mathbf{x}_\tau^k)$
    Let $\mathbf{x}_k = \mathbf{x}_{t_k+1}^k$ and update $t_k$
**Output**: $\mathbf{x}_K$

---

Indeed, for a polyhedral constrained convex polynomial, we can have a tighter result, as shown below.

**Proposition 4.** *[27] Let $F(\mathbf{x})$ be a convex polynomial function on $\mathbb{R}^d$ with degree $m$. If $P \subset \mathbb{R}^d$ is a polyhedral set, then the problem $\min_{\mathbf{x} \in P} F(\mathbf{x})$ admits a global error bound: $\forall \mathbf{x} \in P$ there exists $0 < c < \infty$ such that*

$$D(\mathbf{x}, \Omega_*) \le c \left[ (F(\mathbf{x}) - F_*) + (F(\mathbf{x}) - F_*)^{\frac{1}{m}} \right]. \tag{6}$$

From the global error bound (6), one can easily derive the HEB condition (3). As an example, an $\ell_1$ constrained $\ell_p$ norm regression below [19] satisfies the HEB condition (3) with $\theta = \frac{1}{p}$:

$$\min_{\|\mathbf{x}\|_1 \le s} F(\mathbf{x}) \triangleq \frac{1}{n} \sum_{i=1}^n (\mathbf{a}_i^\top \mathbf{x} - b_i)^p, \quad p \in 2\mathbb{N}. \tag{7}$$

Many previous papers have considered a family of structured smooth composite functions $F(\mathbf{x}) = h(A\mathbf{x}) + g(\mathbf{x})$, where $g(\mathbf{x})$ is a polyhedral function and $h(\cdot)$ is a smooth and strongly convex function on any compact set. Suppose the optimal set of the above problem is non-empty and compact (e.g., the function is coercive) so is the sublevel set $\mathcal{S}_\xi$, and it can been shown that such a function satisfies HEB with $\theta = 1/2$ on any sublevel set $\mathcal{S}_\xi$ [15, Theorem 10]. Examples of $h(\mathbf{u})$ include logistic loss $h(\mathbf{u}) = \sum_i \log(1 + \exp(-u_i))$ and square loss $h(\mathbf{u}) = \|\mathbf{u}\|_2^2$.

Finally, we note that there exist problems that admit HEB with $\theta > 1/2$. A trivial example is given by $F(\mathbf{x}) = \frac{1}{2}\|\mathbf{x}\|_2^2 + \|\mathbf{x}\|_p^p$ with $p \in [1, 2)$, which satisfies HEB with $\theta = 1/p \in (1/2, 1]$. An interesting non-trivial family of problems is that $f(\mathbf{x}) = 0$ and $g(\mathbf{x})$ is a piece-wise linear functions according to Proposition 3. PG or APG applied to such family of problems is closely related to proximal point algorithm [20]. Explorations of such algorithmic connection is not the focus of this paper.

## 3 PG and restarting APG under HEB

As a warm-up and motivation of the major contribution presented in next section, we present a convergence result of PG and a restarting APG under the HEB condition. The analysis is mostly straightforward and is included in the supplement. We first present a result of PG using the update (4).

**Theorem 1.** *Suppose $F(\mathbf{x}_0) - F_* \le \epsilon_0$ and $F(\mathbf{x})$ satisfies HEB on $\mathcal{S}_{\epsilon_0}$. The iteration complexity of PG with option I (which returns the last solution, see the supplementary material) for achieving $F(\mathbf{x}_t) - F_* \le \epsilon$ is $O(c^2 L \epsilon_0^{2\theta-1})$ if $\theta > 1/2$, and is $O(\max\{\frac{c^2 L}{\epsilon^{1-2\theta}}, c^2 L \log(\frac{\epsilon_0}{\epsilon})\})$ if $\theta \le 1/2$.*

Next, we show that APG can be made adaptive to HEB by periodically restarting given $c$ and $\theta$. This is similar to [15] under the QGC. The steps of restarting APG (rAPG) are presented in Algorithm 2, where we employ the simplest variant of APG.

**Theorem 2.** *Suppose $F(\mathbf{x}_0) - F_* \le \epsilon_0$ and $F(\mathbf{x})$ satisfies HEB on $\mathcal{S}_{\epsilon_0}$. By running Algorithm 2 with $K = \lceil \log_2 \frac{\epsilon_0}{\epsilon} \rceil$ and $t_k = \lceil 2c\sqrt{L}\epsilon_{k-1}^{\theta-1/2} \rceil$, we have $F(\mathbf{x}_K) - F_* \le \epsilon$. The iteration complexity of rAPG is $O(c\sqrt{L}\epsilon_0^{1/2-\theta})$ if $\theta > 1/2$, and if $\theta \le 1/2$ it is $O(\max\{\frac{c\sqrt{L}}{\epsilon^{1/2-\theta}}, c\sqrt{L}\log(\frac{\epsilon_0}{\epsilon})\})$.*

From Algorithm 2, we can see that rAPG requires the knowledge of $c$ besides $\theta$ to restart APG. However, for many problems of interest, the value of $c$ is unknown, which makes rAPG impractical. To address this issue, we propose to use the magnitude of the proximal gradient as a measure for restart and termination. It is worth mentioning the difference between the development in this paper and previous studies. Previous work [16, 11] have considered strongly convex optimization

problems where the strong convexity parameter is unknown, where they also use the magnitude of the proximal gradient as a measure for restart and termination. However, in order to achieve faster convergence under the HEB condition without the strong convexity, we have to introduce a novel technique of adaptive regularization that adapts to the HEB. With a novel synthesis of the adaptive regularization and a conditional restarting that searchs for the $c$, we are able to develop practical adaptive accelerated gradient methods. We also notice a recent work [6] that proposed unconditional restarted accelerated gradient methods under QGC. Their restart of APG/FISTA does not involve evaluation of the gradient or the objective value but rather depends on a restarting frequency parameter and a convex combination parameter for computing the restarting solution, which can be set based on a rough estimate of the strong convexity parameter. As a result, their linear convergence (established for distance of solutions to the optimal set) heavily depends on the rough estimate of the strong convexity parameter.

Before diving into the details of the proposed algorithm, we will first present a variant of PG as a baseline for comparison motivated by [18] for smooth problems, which enjoys a faster convergence than the vanilla PG in terms of the proximal gradient's norm. The idea is to return a solution that achieves the minimum magnitude of the proximal gradient, i.e., $\min_{1 \leq \tau \leq t} \|G(\mathbf{x}_\tau)\|_2$. The convergence of $\min_{1 \leq \tau \leq t} \|G(\mathbf{x}_\tau)\|_2$ under HEB is presented in the following theorem.

**Theorem 3.** *Suppose $F(\mathbf{x}_0) - F_* \leq \epsilon_0$ and $F(\mathbf{x})$ satisfies HEB on $\mathcal{S}_{\epsilon_0}$. The iteration complexity of PG (option II, which returns the solution with historically minimal proximal gradient, see the supplementary material) for achieving $\min_{1 \leq \tau \leq t} \|G(\mathbf{x}_\tau)\|_2 \leq \epsilon$, is $O(c^{\frac{1}{1-\theta}} L \max\{1/\epsilon^{\frac{1-2\theta}{1-\theta}}, \log(\frac{\epsilon_0}{\epsilon})\})$ if $\theta \leq 1/2$, and is $O(c^2 L \epsilon_0^{2\theta-1})$ if $\theta > 1/2$.*

The final theorem in this section summarizes an $o(1/t)$ convergence result of PG for minimizing a proper, coercive, convex, lower semicontinuous and semi-algebraic function, which could be interesting of its own.

**Theorem 4.** *Let $F(\mathbf{x})$ be a proper, coercive, convex, lower semicontinuous and semi-algebraic functions. Then PG (with option I and option II) converges at a speed of $o(1/t)$ for $F(\mathbf{x}) - F_*$ and $G(\mathbf{x})$, respectively, where $t$ is the total number of iterations.*

**Remark:** This can be easily proved by combining Proposition 1 and Theorems 1, 3.

## 4 Adaptive Accelerated Gradient Converging Methods

We first present a key lemma for our development that serves the foundation of the adaptive regularization and conditional restarting.

**Lemma 1.** *Assume $F(\mathbf{x})$ satisfies HEB for any $\mathbf{x} \in \mathcal{S}_\xi$ with $\theta \in (0, 1]$. If $\theta \in (0, 1/2]$, then for any $\mathbf{x} \in \mathcal{S}_\xi$, we have $D(\mathbf{x}, \Omega_*) \leq \frac{2}{L} \|G(\mathbf{x})\|_2 + c^{\frac{1}{1-\theta}} 2^{\frac{\theta}{1-\theta}} \|G(\mathbf{x})\|_2^{\frac{\theta}{1-\theta}}$. If $\theta \in (1/2, 1]$, then for any $\mathbf{x} \in \mathcal{S}_\xi$, we have $D(\mathbf{x}, \Omega_*) \leq \left(\frac{2}{L} + 2c^2 \xi^{2\theta-1}\right) \|G(\mathbf{x})\|_2$.*

A building block of the proposed algorithm is to solve a problem of the following style by employing the Algorithm 1 (i.e., Nesterov's ADG):

$$F_\delta(\mathbf{x}) = F(\mathbf{x}) + \frac{\delta}{2} \|\mathbf{x} - \mathbf{x}_0\|_2^2 = f(\mathbf{x}) + g(\mathbf{x}) + \frac{\delta}{2} \|\mathbf{x} - \mathbf{x}_0\|_2^2, \tag{8}$$

which consists of a $L$-smooth function $f(\mathbf{x})$ and a $\delta$-strongly convex function $g_\delta(\mathbf{x}) = g(\mathbf{x}) + \frac{\delta}{2} \|\mathbf{x} - \mathbf{x}_0\|_2^2$. A key result for our development of conditional restarting is the following theorem for each call of Algorithm 1 for solving the above problem.

**Theorem 5.** *By running the Algorithm 1 for minimizing $f(\mathbf{x}) + g_\delta(\mathbf{x})$ with an initial solution $\mathbf{x}_0$, for $t \geq \sqrt{\frac{L}{2\delta}} \log\left(\frac{L}{\delta}\right)$ we have*

$$\|G(\mathbf{x}_{t+1})\|_2 \leq \sqrt{L(L+\delta)} \|\mathbf{x}_0 - \mathbf{x}_*\|_2 \left[1 + \sqrt{\delta/(2L)}\right]^{-t} + 2\sqrt{2}\delta \|\mathbf{x}_0 - \mathbf{x}_*\|_2.$$

*where $\mathbf{x}_*$ is any optimal solution to the original problem.*

Finally, we present the proposed adaptive accelerated gradient converging (adaAGC) method for solving the smooth composite optimization in Algorithm 3 and prove the main theorem of this section.

**Algorithm 3:** adaAGC for solving (1)

---

**Input**: $\mathbf{x}_0 \in \Omega$ and $c_0$ and $\gamma > 1$

Let $c_e = c_0$ and $\varepsilon_0 = \|G(\mathbf{x}_0)\|_2$

**for** $k = 1, \ldots, K$ **do**

    **for** $s = 1, \ldots,$ **do**

        Let $\delta_k$ be given in (9) and $g_{\delta_k}(\mathbf{x}) = g(\mathbf{x}) + \frac{\delta_k}{2}\|\mathbf{x} - \mathbf{x}_{k-1}\|_2^2$

        $A_0 = 0, \mathbf{v}_0 = \mathbf{x}_{k-1}, \mathbf{x}_0^k = \mathbf{x}_{k-1}$

        **for** $t = 0, \ldots$ **do**

            Let $a_{t+1}$ be the root of $\frac{a^2}{A_t + a} = 2\frac{1 + \delta_k A_t}{L}$

            Set $A_{t+1} = A_t + a_{t+1}$

            Set $\mathbf{y}_t = \frac{A_t}{A_{t+1}}\mathbf{x}_t^k + \frac{a_{t+1}}{A_{t+1}}\mathbf{v}_t$

            Compute $\mathbf{x}_{t+1}^k = P_{g_{\delta_k}/L}(\mathbf{y}_t - \nabla f(\mathbf{y}_t)/L)$

            Compute $\mathbf{v}_{t+1} = \arg\min_{\mathbf{x}} \frac{1}{2}\|\mathbf{x} - \mathbf{x}_{k-1}\|_2^2 + \sum_{\tau=1}^{t+1} a_\tau \nabla f(\mathbf{x}_\tau^k)^\top \mathbf{x} + A_{t+1} g_{\delta_k}(\mathbf{x})$

            **if** $\|G(\mathbf{x}_{t+1}^k)\|_2 \le \varepsilon_{k-1}/2$ **then**

                let $\mathbf{x}_k = \mathbf{x}_{t+1}^k$ and $\varepsilon_k = \varepsilon_{k-1}/2$     `// step S1`

                break the enclosing two for loops

            **if** $\tau = \lceil\sqrt{\frac{2L}{\delta_k}}\log\frac{\sqrt{L(L+\delta_k)}}{\delta_k}\rceil$ **then**     `// condition (*)`

                let $c_e = \gamma c_e$ and break the enclosing for loop     `// step S2`

**Output**: $\mathbf{x}_K$

The adaAGC runs with multiple stages ($k = 1, \ldots, K$). We start with an initial guess $c_0$ of the parameter $c$ in the HEB. With the current guess $c_e$ of $c$, at the $k$-th stage adaAGC employs ADG to solve a problem of (8) with an adaptive regularization parameter $\delta_k$ being

$$\delta_k = \begin{cases} \min\left(\frac{L}{32}, \frac{\varepsilon_{k-1}^{\frac{1-2\theta}{1-\theta}}}{16c_e^{1/(1-\theta)}2^{\frac{\theta}{1-\theta}}}\right) & \text{if } \theta \in (0, 1/2] \\ \min\left(\frac{L}{32}, \frac{1}{32c_e^2\epsilon_0^{2\theta-1}}\right) & \text{if } \theta \in (1/2, 1] \end{cases} \tag{9}$$

The condition (*) specifies the condition for restarting with an increased value of $c_e$. When the flow enters step S2 before step S1 for each $s$, it means that the current guess $c_e$ is not sufficiently large according to Theorem 5 and Lemma 1, then we increase $c_e$ and repeat the same process (next iteration for $s$). We refer to this machinery as conditional restarting. We present the main result of this section in the following theorem.

**Theorem 6.** *Suppose* $F(\mathbf{x}_0) - F_* \le \epsilon_0$, $F(\mathbf{x})$ *satisfies HEB on* $\mathcal{S}_{\epsilon_0}$ *and* $c_0 \le c$. *Let* $\varepsilon_0 = \|G(\mathbf{x}_0)\|_2$, $K = \lceil\log_2(\frac{\varepsilon_0}{\epsilon})\rceil$, $p = (1 - 2\theta)/(1 - \theta)$ *for* $\theta \in (0, 1/2]$. *The iteration complexity of Algorithm 3 for having* $\|G(\mathbf{x}_K)\|_2 \le \epsilon$ *is* $\widetilde{O}\left(\sqrt{L}c^{\frac{1}{2(1-\theta)}}\max(\frac{1}{\epsilon^{p/2}}, \log(\varepsilon_0/\epsilon))\right)$ *if* $\theta \in (0, 1/2]$, *and* $\widetilde{O}(\sqrt{L}c\epsilon_0^{\theta-1/2})$ *if* $\theta \in (1/2, 1]$, *where* $\widetilde{O}(\cdot)$ *suppresses a log term depending on* $c, c_0, L, \gamma$.

We sketch the idea of the proof here: for each $k$, we can bound the number of cycles (indexd by $s$ in the algorithm) in order to enter step S1 denoted by $s_k$. We can bound $s_k \le \log_\gamma(c/c_0) + 1$ and then total number of iterations across all stages is bounded by $\sum_{k=1}^K s_k t_k$ where $t_k = \lceil\sqrt{\frac{2L}{\delta_k}}\log\frac{\sqrt{L(L+\delta_k)}}{\delta_k}\rceil$.

Before ending this section, we would like to remark that if the smoothness parameter $L$ is unknown, one can also employ the backtracking technique pairing with each update to search for $L$ [17].

## 4.1 Convergence of Objective Gap

In this subsection, we show that the convergence of the proximal gradient also implies the convergence of the objective gap $F(\mathbf{x}) - F_*$ for certain subclasses of the general problems that we have considered. Our first result applies to the case when $F(\mathbf{x})$ satisfies the HEB with $\theta \in (0, 1)$ and the nonsmooth part $g(\mathbf{x})$ is absent, i.e., $F(\mathbf{x}) = f(\mathbf{x})$. In this case, we can establish the convergence of the objective gap, since the objective gap can be bounded by a function of the magnitude of gradient,

i.e., $f(\mathbf{x}) - f_* \leq c^{1/(1-\theta)} \|\nabla f(\mathbf{x})\|_2^{1/(1-\theta)}$ (c.f. the proof of Lemma 2 in the supplement). One can easily prove the following result.

**Theorem 7.** *Assume $F(\mathbf{x}) = f(\mathbf{x})$ and the same conditions in Theorem 6 hold. The iteration complexity of Algorithm 3 for having $F(\mathbf{x}_K) - F(\mathbf{x}_*) \leq \epsilon$ is $\widetilde{O}\left(\sqrt{L}c \max(\frac{1}{\epsilon^{1/2-\theta}}, \log(\varepsilon_0/\epsilon))\right)$ if $\theta \in (0, 1/2]$, and $\widetilde{O}(\sqrt{L}c\epsilon_0^{\theta-1/2})$ if $\theta \in (1/2, 1)$, where $\widetilde{O}(\cdot)$ suppresses a log term depending on $c, c_0, L, \gamma$.*

**Remark** Note that the above iteration complexity of adaAGC is the same as that of rAPG (shown in Table 1), where the later is established under the knowledge of $c$.

Our second result applies to a subclass of the general problems where either $g(\mathbf{x})$ or $f(\mathbf{x})$ is $\mu$-strongly convex or $F(\mathbf{x}) = f(\mathbf{x}) + g(\mathbf{x})$, where $f(\mathbf{x}) = h(A\mathbf{x})$ with $h(\cdot)$ being a strongly convex function and $g(\mathbf{x})$ is the indicator function of a polyhedral set $\Omega = \{\mathbf{x} : C\mathbf{x} \leq b\}$. Examples include square loss minimization under an $\ell_1$ or $\ell_\infty$ constraint [15, Theorem 8]. It has been shown that in the last case, for any $\mathbf{x} \in \text{dom}(F)$, there exists $\mu > 0$ such that

$$f(\mathbf{x}_*) \geq f(\mathbf{x}) + \nabla f(\mathbf{x})^\top (\mathbf{x}_* - \mathbf{x}) + \frac{\mu}{2}\|\mathbf{x} - \mathbf{x}_*\|_2^2, \tag{10}$$

where $\mathbf{x}_*$ is the closest optimal solution to $\mathbf{x}$, and the HEB condition of $F(\mathbf{x})$ with $\theta = 1/2$ and $c = \sqrt{2/\mu}$ holds [15, Theorem 1]. In the three cases mentioned above, we can establish that $F(\mathbf{x}^+) - F_* \leq O(1/\mu)\|G(\mathbf{x})\|_2^2$, where $\mathbf{x}^+ = P_{g/L}(\mathbf{x} - \nabla f(\mathbf{x})/L)$, and the following result.

**Theorem 8.** *Assume $f(\mathbf{x})$ or $g(\mathbf{x})$ is $\mu$-strongly convex, or $f(\mathbf{x}) = h(A\mathbf{x})$ and $g(\mathbf{x})$ is the indicator function of a polyhedral set such that (10) holds for some $\mu > 0$, and other conditions in Theorem 6 hold. The iteration complexity of Algorithm 3 for having $F(\mathbf{x}_K^+) - F(\mathbf{x}_*) \leq \epsilon$ is $\widetilde{O}\left(\sqrt{L/\mu}\log(\varepsilon_0/\sqrt{\mu\epsilon})\right)$, where $\widetilde{O}(\cdot)$ suppresses a log term depending on $\mu, c_0, L, \gamma$.*

## 5 Applications and Experiments

In this section, we present some applications of our theorems and algorithms in machine learning. In particular, we consider the regularized problems with a smooth loss:

$$\min_{\mathbf{x} \in \mathbb{R}^d} \frac{1}{n} \sum_{i=1}^{n} \ell(\mathbf{x}^\top \mathbf{a}_i, b_i) + \lambda R(\mathbf{x}), \tag{11}$$

where $(\mathbf{a}_i, b_i), i = 1, \ldots, n$ denote a set of training examples, $R(\mathbf{x})$ could be the $\ell_1$ norm $\|\mathbf{x}\|_1$, the $\ell_\infty$ norm $\|\mathbf{x}\|_\infty$, or a huber norm [28], or the $\ell_{1,p}$ norm $\sum_{k=1}^{K}\|\mathbf{x}_k\|_p$, where $k$ is the $k$-th component vector of $\mathbf{x}$. Next, we present several results about the HEB condition to cover a broad family of loss functions that enjoy the faster convergence of adaAGC.

**Corollary 1.** *Assume the loss function $\ell(z, b)$ is nonnegative, convex, smooth and piecewise quadratic, then the problems in (11) with $\ell_1$ norm, $\ell_\infty$ norm, Huber norm and $\ell_{1,\infty}$ norm regularization satisfy the HEB condition with $\theta = 1/2$ on any sublevel set $S_\xi$ with $\xi > 0$. Hence adaAGC has a global linear convergence in terms of the proximal gradient's norm and a square root dependence on the condition number.*

**Remark:** The above corollary follows directly from Proposition 2 and Theorem 6. If the loss function is a logistic loss and the regularizer is a polyhedral function (e.g., $\ell_1, \ell_\infty$ and $\ell_{1,\infty}$ norm), we can prove the same result. Examples of convex, smooth and piecewise convex quadratic loss functions include: square loss: $\ell(z, b) = (z - b)^2$ for $b \in \mathbb{R}$; squared hinge loss: $\ell(z, b) = \max(0, 1 - bz)^2$ for $b \in \{1, -1\}$; and huber loss: $\ell(z, b) = \rho(|z - b| - \frac{\rho}{2})$ if $|z - b| > \rho$, and $\ell(z, b) = (z - b)^2/2$ if $|z - b| \leq \rho$, for $b \in \mathbb{R}$.

**Experimental Results** We conduct some experiments to demonstrate the effectiveness of adaAGC for solving problems of type (1). Specifically, we compare adaAGC, PG with option II that returns the solution with historically minimal proximal gradient, FISTA, unconditional restarting FISTA (urFISTA) [6] for optimizing the squared hinge loss (classification), square loss (regression), huber loss (with $\rho = 1$) (regression) with $\ell_1$ and $\ell_\infty$ regularization, which are cases of (11), and we also consider the $\ell_1$ constrained $\ell_p$ norm regression (7) with varying $p$. We use three datasets from the LibSVM website [5], which are splice ($n = 1000, d = 60$) for classification, and bodyfat

Table 2: squared hinge loss with $\ell_1$ norm (left) and $\ell_\infty$ norm (right) regularization on splice data

| Algorithm | $\epsilon = 10^{-4}$ | $\epsilon = 10^{-5}$ | $\epsilon = 10^{-6}$ | $\epsilon = 10^{-7}$ | $\epsilon = 10^{-4}$ | $\epsilon = 10^{-5}$ | $\epsilon = 10^{-6}$ | $\epsilon = 10^{-7}$ |
|---|---|---|---|---|---|---|---|---|
| PG | 2040 | 2040 | 2040 | 2040 | 3514 | 3724 | 3724 | 3724 |
| FISTA | **1289** | **1289** | **1289** | **1289** | 5526 | 5526 | 5526 | 5526 |
| urFISTA | 1666 | 2371 | 2601 | 3480 | **1674** | **2379** | 2605 | 3488 |
| adaAGC | 1410 | 1410 | 1410 | 1410 | 2382 | 2382 | **2382** | **2382** |
| FISTA > adaAGC > PG > urFISTA | | | | adaAGC > urFISTA > PG > FISTA | | | | |

Table 3: square loss with $\ell_1$ norm (left) and $\ell_\infty$ norm (right) regularization on cpusmall data

| Algorithm | $\epsilon = 10^{-4}$ | $\epsilon = 10^{-5}$ | $\epsilon = 10^{-6}$ | $\epsilon = 10^{-7}$ | $\epsilon = 10^{-4}$ | $\epsilon = 10^{-5}$ | $\epsilon = 10^{-6}$ | $\epsilon = 10^{-7}$ |
|---|---|---|---|---|---|---|---|---|
| PG | 109298 | 159908 | 170915 | 170915 | 139505 | 204120 | 210874 | 210874 |
| FISTA | **6781** | 16387 | 23779 | 23779 | **6610** | 16418 | 20082 | 20082 |
| urFISTA | 18278 | 26706 | 35173 | 43603 | 18276 | 26704 | 35169 | 43601 |
| adaAGC | 9571 | **12623** | **13575** | **13575** | 9881 | **13033** | **13632** | **13632** |
| adaAGC > FISTA > urFISTA > PG | | | | adaAGC > FISTA > urFISTA > PG | | | | |

Table 4: $\ell_1$ regularized huber loss (left) and $\ell_1$ constrained square loss (right) on bodyfat data

| Algorithm | $\epsilon = 10^{-4}$ | $\epsilon = 10^{-5}$ | $\epsilon = 10^{-6}$ | $\epsilon = 10^{-7}$ | $\epsilon = 10^{-4}$ | $\epsilon = 10^{-5}$ | $\epsilon = 10^{-6}$ | $\epsilon = 10^{-7}$ |
|---|---|---|---|---|---|---|---|---|
| PG | 258723 | 423181 | 602043 | 681488 | 1006880 | 1768482 | 2530085 | 2632578 |
| FISTA | **6630** | 25020 | 74416 | 124261 | **15805** | 66319 | 180977 | 181176 |
| urFISTA | 6855 | **12662** | **17994** | **23933** | 138359 | 235081 | 331203 | 426341 |
| adaAGC | 16976 | 16980 | 23844 | 25697 | 23054 | **33818** | **44582** | **48127** |
| urFISTA > adaAGC > FISTA > PG | | | | adaAGC > FISTA > urFISTA > PG | | | | |

Table 5: $\ell_1$ constrained $\ell_p$ norm regression on bodyfat data ($\epsilon = 10^{-3}$)

| Algorithm | $p = 2$ | $p = 4$ | $p = 6$ | $p = 8$ |
|---|---|---|---|---|
| PG | 250869 (1) | 979401 (3.90) | 1559753 (6.22) | 4015665 (16.00) |
| adaAGC | **8710 (1)** | **17494 (2.0)** | **22481 (2.58)** | **33081 (3.80)** |

($n = 252, d = 14$), cpusmall ($n = 8192, d = 12$) for regression. For problems covered by (11), we fix $\lambda = \frac{1}{n}$, and the parameter $s$ in (7) is set to $s = 100$.

We use the backtracking in PG, adaAGC and FISTA to search for the smoothness parameter. In adaAGC, we set $c_0 = 2, \gamma = 2$ for the $\ell_1$ constrained $\ell_p$ norm regression and $c_0 = 10, \gamma = 2$ for the rest problems. For fairness, for urFISTA and adaAGC, we use the same initial estimate of unknown parameter (i.e., $c$). Each algorithm starts at the same initial point, which is set to be zero, and we stop each algorithm when the norm of its proximal gradient is less than a prescribed threshold $\epsilon$ and report the total number of proximal mappings. The results are presented in the Tables 2–5. It indicates that adaAGC converges faster than PG and FISTA (except for solving squared hinge loss with $\ell_1$ norm regularization) when $\epsilon$ is very small, which is consistent with the theoretical results. Note that urFISTA sometimes has better performance than adaAGC but is worse than adaAGC in most cases. It is notable that for some problems (see Table 2) the number of proximal mappings is the same value for achieving different precision $\epsilon$. This is because that value is the minimum number of proximal mappings such that the magnitude of the proximal gradient suddenly becomes zero. In Table 5, the numbers in parenthesis indicate the increasing factor in the number of proximal mappings compared to the base case $p = 2$, which show that increasing factors of adaAGC are approximately the square root of that of PG and thus are consistent with our theory.

## 6 Conclusions

In this paper, we have considered smooth composite optimization problems under a general Hölderian error bound condition. We have established adaptive iteration complexity to the Hölderian error bound condition of proximal gradient and accelerated proximal gradient methods. To eliminate the dependence on the unknown parameter in the error bound condition and enjoy the faster convergence of accelerated proximal gradient method, we have developed a novel parameter-free adaptive accelerated gradient converging method using the magnitude of the (proximal) gradient as a measure for restart and termination. We have also considered a broad family of norm regularized problems in machine learning and showed faster convergence of the proposed adaptive accelerated gradient converging method.

**Acknowledgments** We thank the anonymous reviewers for their helpful comments. M. Liu and T. Yang are partially supported by National Science Foundation (IIS-1463988, IIS-1545995).

## Footnotes

[1] For the moment, we neglect the constant factor.

[2] It can be relaxed to a fixed domain as done in this work.

[3] When $\theta > 1/2$, all algorithms can converge in finite steps.

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
