[Supplementary Material · supplement.pdf]

# Supplementary Material for "Adaptive Accelerated Gradient Converging Method under Hölderian Error Bound Condition"

**Mingrui Liu, Tianbao Yang**
Department of Computer Science
The University of Iowa, Iowa City, IA 52242
`mingrui-liu, tianbao-yang@uiowa.edu`

We first present the two options of PG.

---

**Algorithm:** PG
**Input**: $\mathbf{x}_1 \in \Omega$
**for** $\tau = 1, \ldots, t$ **do**
$\quad \lfloor \quad \mathbf{x}_{\tau+1} = P_{g/L}(\mathbf{x}_\tau - \nabla f(\mathbf{x}_\tau)/L)$
Option I: return $\mathbf{x}_{t+1}$
Option II: return $\mathbf{x}_k$ s.t. $G(\mathbf{x}_k) = \min_\tau \|G(\mathbf{x}_\tau)\|_2$

---

## 1 Definitions

We introduce two definitions that are mentioned in section 2: semi-algebraic set and semi-algebraic function [2].

**Definition 2.** *A subset $S \subset \mathbb{R}^d$ is called a real semi-algebraic set if there exist a finite number of real polynomial functions $g_{ij}, h_{ij} : \mathbb{R}^d \to \mathbb{R}$ such that*

$$S = \cup_{j=1}^p \cap_{i=1}^q \{\mathbf{u} \in \mathbb{R}^d; g_{ij}(\mathbf{u}) = 0 \text{ and } h_{ij}(\mathbf{u}) \leq 0\}.$$

**Definition 3.** *A function $F(\mathbf{x})$ is called a semi-algebraic function if its graph $\{(\mathbf{u}, s) \in \mathbb{R}^{d+1} : F(\mathbf{u}) = s\}$ is a semi-algebraic set.*

## 2 Propositions

We introduce some results that are useful for our further analysis.

**Proposition 5.** *[7] Assume $f(\mathbf{x})$ is $L$-smooth and $g(\mathbf{x})$ is $\alpha$-strongly convex. Let ADG (Algorithm 1) run for $t = 0, \ldots, T$ iterations. Then for any $\mathbf{x}$ we have*

$$F(\mathbf{x}_{T+1}) - F(\mathbf{x}) \leq \frac{L}{2} \|\mathbf{x}_0 - \mathbf{x}\|_2^2 \left( \frac{1}{1 + \sqrt{\alpha/2L}} \right)^{2T}.$$

**Proposition 6.** *[1, Lemma 2.3] Let $F(\mathbf{x}) = f(\mathbf{x}) + g(\mathbf{x})$. Assume $f(\mathbf{x})$ is $L$-smooth. For any $\mathbf{x}, \mathbf{y}$ and $\eta \leq 1/L$, we have*

$$F(\mathbf{y}_\eta^+) \leq F(\mathbf{x}) + G_\eta(\mathbf{y})^\top (\mathbf{y} - \mathbf{x}) - \frac{\eta}{2} \|G_\eta(\mathbf{y})\|_2^2.$$

**Proposition 7.** *[1, Theorem 3.1] Consider PG with option I, whose update formula is*

$$\mathbf{x}_{t+1} = P_{\eta g}(\mathbf{x}_t - \eta \nabla f(\mathbf{x}_t)). \tag{11}$$

*Let (11) run for $t = 1, \ldots, T$ iterations with $\eta \leq 1/L$, we have*

$$F(\mathbf{x}_{T+1}) - F_* \leq \frac{D(\mathbf{x}_1, \Omega_*)^2}{2\eta T}.$$

**Proposition 8.** *[8] Consider one specific variant of APG, whose update formula is*

$$\begin{cases} \mathbf{y}_t = \mathbf{x}_t + \beta_t(\mathbf{x}_t - \mathbf{x}_{t-1}), \\ \mathbf{x}_{t+1} = P_{\eta g}(\mathbf{y}_t - \eta \nabla f(\mathbf{y}_t)), \end{cases} \tag{12}$$

*where $\eta \leq 1/L$ and $\beta_t = \frac{t-1}{t+2}$. Let (12) run for $t = 1, \ldots, T$ iterations with $\eta \leq 1/L$ and $\mathbf{x}_0 = \mathbf{x}_1$, we have*

$$F(\mathbf{x}_{T+1}) - F_* \leq \frac{2D(\mathbf{x}_1, \Omega_*)^2}{\eta(T+1)^2}.$$

**Proposition 9.** *[5, Theorem 1] Assume $f(\mathbf{x})$ is $L$-smooth and $\alpha$-strongly convex. Let (12) run for $t = 1, \ldots, T$ with $\eta = 1/L$, $\beta_t = \frac{\sqrt{L} - \sqrt{\alpha}}{\sqrt{L} + \sqrt{\alpha}}$ and $\mathbf{x}_0 = \mathbf{x}_1$, we have for any $\mathbf{x}$*

$$F(\mathbf{x}_{T+1}) - F(\mathbf{x}) \leq \left(1 - \sqrt{\frac{\alpha}{L}}\right)^T \left[F(\mathbf{x}_0) - F(\mathbf{x}) + \frac{\alpha}{2}\|\mathbf{x}_0 - \mathbf{x}\|_2^2\right].$$

**Proposition 10.** *[3, Theorem 5 in v3] Let $f : H \to (-\infty, +\infty]$ be a proper, convex and lower semi-continuous with $\min f = f_*$. Let $r_0 > 0$, $\varphi \in \{\varphi \in C^0[0, r_0) \cap C^1(0, r_0), \varphi(0) = 0, \varphi \text{ is concave}, \varphi > 0\}$, $c > 0, \rho > 0$, and $\bar{x} \in \arg\min f$. If $s\varphi'(s) \geq c\varphi(s)$ for all $s \in (0, r_0)$, and $\varphi(f(x) - f_*) \geq D(x, \arg\min f)$ for all $x \in [0 < f < r_0] \cap B(\bar{x}, \rho)$, then $\varphi'(f(x) - f_*)\|\partial f(x)\|_2 \geq c$ for all $x \in [0 < f < r_0] \cap B(\bar{x}, \rho)$.*

The following proposition is a rephrase of Theorem 3.5 in [4].

**Proposition 11.** *If $f$ is $L$-smooth and convex, $g$ is proper, convex and lower semi-continuous, $F(\mathbf{x}) = f(\mathbf{x}) + g(\mathbf{x})$, $\eta > 0$, and define*

$$P_{\eta F}(\mathbf{x}) = \arg\min_{\mathbf{u}} \frac{1}{2}\|\mathbf{u} - \mathbf{x}\|_2^2 + \eta F(\mathbf{u}).$$

*Then the following inequality holds:*

$$\left\|\frac{1}{\eta}(\mathbf{x} - P_{\eta F}(\mathbf{x}))\right\|_2 \leq (1 + L\eta)\|G_\eta(\mathbf{x})\|_2.$$

## 3 Lemmas and Corollaries

**Lemma 2.** *If $f(\mathbf{x})$ satisfies the HEB on $\mathbf{x} \in \mathcal{S}_\xi$ with $\theta \in (0, 1]$, i.e., there exists $c > 0$ such that for any $\mathbf{x} \in \mathcal{S}_\xi$, we have*

$$D(\mathbf{x}, \Omega_*) \leq c(f(\mathbf{x}) - f_*)^\theta.$$

*If $\theta \in (0, 1)$, then for any $\mathbf{x} \in \mathcal{S}_\xi$,*

$$D(\mathbf{x}, \Omega_*) \leq c^{\frac{1}{1-\theta}}\|\partial f(\mathbf{x})\|_2^{\frac{\theta}{1-\theta}}.$$

*If $\theta = 1$, then for any $\mathbf{x} \in \mathcal{S}_\xi$,*

$$D(\mathbf{x}, \Omega_*) \leq c^2 \xi \|\partial f(\mathbf{x})\|_2.$$

*Proof.* The conclusion is trivial if $\mathbf{x} \in \Omega_*$. Otherwise, the proof follows Proposition 10. In particular, if we define $\varphi(s) = cs^\theta$, then $D(\mathbf{x}, \Omega_*) \leq \varphi(f(\mathbf{x}) - f_*)$ for any $\mathbf{x} \in \{\mathbf{x} : 0 < f(\mathbf{x}) - f_* \leq \xi\}$ and $\varphi$ satisfies $s\varphi'(s) \geq \theta\varphi(s)$. By Proposition 10, we have

$$\varphi'(f(\mathbf{x}) - f_*)\|\partial f(\mathbf{x})\|_2 \geq \theta,$$

i.e.,

$$c\|\partial f(\mathbf{x})\|_2 \geq (f(\mathbf{x}) - f_*)^{1-\theta}. \tag{13}$$

When $\theta = 1$, we have $\|\partial f(\mathbf{x})\|_2 \geq 1/c$ for $\mathbf{x} \notin \Omega_*$. As a result, when $\theta \in (0, 1)$,

$$D(\mathbf{x}, \Omega_*) \leq c(f(\mathbf{x}) - f_*)^\theta \leq c^{\frac{1}{1-\theta}} \|\partial f(\mathbf{x})\|_2^{\frac{\theta}{1-\theta}}.$$

and when $\theta = 1$,

$$D(\mathbf{x}, \Omega_*) \leq c(f(\mathbf{x}) - f_*) \leq c^2 \xi \|\partial f(\mathbf{x})\|_2.$$

$\square$

**Corollary 2.** *Let $F(\mathbf{x}) = f(\mathbf{x}) + g(\mathbf{x})$. Assume $f(\mathbf{x})$ is $L$-smooth. For any $\mathbf{x}, \mathbf{y}$ and $0 < \eta \leq 1/L$, we have*

$$\frac{\eta}{2} \|G_\eta(\mathbf{y})\|_2^2 \leq F(\mathbf{y}) - F(\mathbf{y}_\eta^+) \leq F(\mathbf{y}) - \min_{\mathbf{x}} F(\mathbf{x}). \tag{14}$$

*Proof.* The proof is immediate by employing the convexity of $F$ and Proposition 6. $\square$

**Lemma 3.** *By running the ADG (Algorithm 1) for minimizing $F_\delta(\mathbf{x}) = f(\mathbf{x}) + g_\delta(\mathbf{x})$ with an initial solution $\mathbf{x}_0$, where $g_\delta(\mathbf{x}) = g(\mathbf{x}) + \frac{\delta}{2}\|\mathbf{x} - \mathbf{x}_0\|_2^2$, then for any $\mathbf{x} \in \mathbb{R}^d$ and $t \geq 0$,*

$$F_\delta(\mathbf{x}_{t+1}) - F_\delta(\mathbf{x}) \leq \frac{L}{2}\|\mathbf{x}_0 - \mathbf{x}\|_2^2 \left[1 + \sqrt{\frac{\delta}{2L}}\right]^{-2t},$$

*and $F(\mathbf{x}_{t+1}) \leq F(\mathbf{x}_0)$. If $t \geq \sqrt{\frac{L}{2\delta}} \log\left(\frac{L}{\delta}\right)$, we have $\|\mathbf{x}_{t+1} - \mathbf{x}_0\|_2 \leq \sqrt{2}\|\mathbf{x}_0 - \mathbf{x}_*\|_2$.*

*Proof.* Applying Proposition 5 to $F_\delta(\mathbf{x})$ yields

$$F(\mathbf{x}_{t+1}) - F(\mathbf{x}) + \frac{\delta}{2}\|\mathbf{x}_{t+1} - \mathbf{x}_0\|_2^2 \leq \frac{\delta}{2}\|\mathbf{x} - \mathbf{x}_0\|_2^2 + \frac{L}{2}\|\mathbf{x}_0 - \mathbf{x}\|_2^2 \left[1 + \sqrt{\frac{\delta}{2L}}\right]^{-2t}. \tag{15}$$

Then $F(\mathbf{x}_{t+1}) - F(\mathbf{x}_0) \leq 0$, and choose $\mathbf{x} = \mathbf{x}_*$ in the inequality (15), where $\mathbf{x}_* \in \Omega_*$, then we have

$$\|\mathbf{x}_{t+1} - \mathbf{x}_0\|_2^2 \leq \|\mathbf{x}_0 - \mathbf{x}_*\|_2^2 + \frac{L}{\delta}\|\mathbf{x}_0 - \mathbf{x}_*\|_2^2 \left[1 + \sqrt{\frac{\delta}{2L}}\right]^{-2t}.$$

Under the condition $t \geq \sqrt{\frac{L}{2\delta}} \log\left(\frac{L}{\delta}\right)$ we have $\|\mathbf{x}_{t+1} - \mathbf{x}_0\|_2 \leq \sqrt{2}\|\mathbf{x}_0 - \mathbf{x}_*\|_2$. $\square$

**Lemma 4** (**Perturbation of a Strongly Convex Problem**). *Let $h(\mathbf{x})$ be a $\sigma$-strongly convex function, $\mathbf{x}_a^*$ and $\mathbf{x}_b^*$ be the optimal solutions to the following problems.*

$$\mathbf{x}_a^* = \min_{\mathbf{x} \in \mathbb{R}^d} \mathbf{a}^\top \mathbf{x} + h(\mathbf{x}).$$

$$\mathbf{x}_b^* = \min_{\mathbf{x} \in \mathbb{R}^d} \mathbf{b}^\top \mathbf{x} + h(\mathbf{x}).$$

*Then*

$$\|\mathbf{x}_a^* - \mathbf{x}_b^*\|_2 \leq \frac{2\|\mathbf{a} - \mathbf{b}\|_2}{\sigma}.$$

*Proof.* Let $H_a(\mathbf{x}) = h(\mathbf{x}) + \mathbf{a}^\top \mathbf{x}$ and $H_b(\mathbf{x}) = h(\mathbf{x}) + \mathbf{b}^\top \mathbf{x}$. By the strong convexity of $h(\mathbf{x})$, we have

$$\frac{\sigma}{2}\|\mathbf{x}_a^* - \mathbf{x}_b^*\|_2^2 \leq H_a(\mathbf{x}_b^*) - H_a(\mathbf{x}_a^*) = H_b(\mathbf{x}_b^*) + (\mathbf{a} - \mathbf{b})^\top \mathbf{x}_b^* - H_b(\mathbf{x}_a^*) - (\mathbf{a} - \mathbf{b})^\top \mathbf{x}_a^*$$

$$\leq (\mathbf{a} - \mathbf{b})^\top (\mathbf{x}_b^* - \mathbf{x}_a^*) \leq \|\mathbf{x}_a^* - \mathbf{x}_b^*\|_2 \|\mathbf{a} - \mathbf{b}\|_2,$$

where we use the fact $H_b(\mathbf{x}_b^*) \leq H_b(\mathbf{x}_a^*)$. From the above inequality, we can get $\|\mathbf{x}_a^* - \mathbf{x}_b^*\|_2 \leq \frac{2\|\mathbf{a}-\mathbf{b}\|_2}{\sigma}$. $\square$

# 4  Proofs

## A  Proof of Theorem 1

*Proof.* Divide the whole FOR loop of PG into $K$ stages, denote $t_k$ by the number of iterations in the $k$-th stage, and denote $\mathbf{x}_k$ by the updated $\mathbf{x}$ at the end of the $k$-th stage, where $k = 1, \ldots K$. Define $\epsilon_k := \frac{\epsilon_0}{2^k}$.

Choose $t_k = \lceil c^2 L \epsilon_{k-1}^{2\theta-1} \rceil$, and we will prove $F(\mathbf{x}_k) - F_* \leq \epsilon_k$ by induction. Suppose $F(\mathbf{x}_{k-1}) - F_* \leq \epsilon_{k-1}$, we have $\mathbf{x}_{k-1} \in \mathcal{S}_{\epsilon_0}$. According to Proposition 7, at the $k$-th stage, we have

$$F(\mathbf{x}_k) - F_* \leq \frac{L\|\mathbf{x}_{k-1} - \mathbf{x}_{k-1}^*\|_2^2}{2t_k},$$

where $\mathbf{x}_{k-1}^* \in \Omega_*$, the closest point to $\mathbf{x}_{k-1}$ in the optimal set. By the HEB condition, we have

$$F(\mathbf{x}_k) - F_* \leq \frac{c^2 L \epsilon_{k-1}^{2\theta}}{2t_k}.$$

Since $t_k \geq c^2 L \epsilon_{k-1}^{2\theta-1}$, we have $F(\mathbf{x}_k) - F_* \leq \epsilon_k$. The total number of iterations is

$$\sum_{k=1}^{K} t_k \leq O(c^2 L \sum_{k=1}^{K} \epsilon_{k-1}^{2\theta-1}).$$

From the above analysis, we see that after each stage, the optimality gap decreases by half, so taking $K = \lceil \log_2 \frac{\epsilon_0}{\epsilon} \rceil$ guarantees $F(\mathbf{x}_k) - F_* \leq \epsilon$.

If $\theta > 1/2$, the iteration complexity is $O(c^2 L \epsilon_0^{2\theta-1})$. To see this, if we plug in the definition of $\epsilon_k$ into the total number of iterations, and we can get $O(c^2 L \epsilon_0^{2\theta-1} \sum_{k=1}^{K} \frac{1}{2^{(2\theta-1)(k-1)}}) = O(c^2 L \epsilon_0^{2\theta-1})$. If $\theta = 1/2$, the iteration complexity is $O(c^2 L \log \frac{\epsilon_0}{\epsilon})$. If $\theta < 1/2$, the iteration complexity is

$$\sum_{k=1}^{K} t_k \leq O(c^2 L \sum_{k=1}^{K} (\frac{\epsilon_0}{2^{k-1}})^{2\theta-1}) = O(c^2 L/\epsilon^{1-2\theta}).$$

$\square$

## B  Proof of Theorem 2

*Proof.* Similar to the proof of Theorem 1, we will prove by induction that $F(\mathbf{x}_k) - F_* \leq \epsilon_k \triangleq \frac{\epsilon_0}{2^k}$. Assume that $F(\mathbf{x}_{k-1}) - F_* \leq \epsilon_{k-1}$. Hence, $\mathbf{x}_{k-1} \in \mathcal{S}_{\epsilon_0}$. Then according to Proposition 8 and the HEB condition, we have

$$F(\mathbf{x}_k) - F_* \leq \frac{2c^2 L \epsilon_{k-1}^{2\theta}}{(t_k+1)^2}.$$

Since $t_k \geq 2c\sqrt{L}\epsilon_{k-1}^{\theta-1/2}$, we have

$$F(\mathbf{x}_k) - F_* \leq \frac{\epsilon_{k-1}}{2} = \epsilon_k.$$

After $K$ stages, we have $F(\mathbf{x}_K) - F_* \leq \epsilon$. The total number of iterations is

$$T_K = \sum_{k=1}^{K} t_k \leq O(c\sqrt{L}\epsilon_{k-1}^{\theta-1/2}).$$

When $\theta > 1/2$, we have $T_K \leq O(c\sqrt{L}\epsilon_0^{\theta-1/2})$. When $\theta \leq 1/2$, we have

$$T_K \leq O\left(\max\{c\sqrt{L}\log(\epsilon_0/\epsilon), c\sqrt{L}/\epsilon^{1/2-\theta}\}\right).$$

$\square$

## C   Proof of Theorem 3

*Proof.* By the update of PG with option II and Corollary 2, we have

$$F(\mathbf{x}_\tau) - F(\mathbf{x}_{\tau+1}) \geq \frac{1}{2L}\|G(\mathbf{x}_\tau)\|_2^2.$$

Let $t = 2j$. Summing over $\tau = j, \ldots, t$ gives

$$F(\mathbf{x}_j) - F(\mathbf{x}_{t+1}) \geq \frac{1}{2L}\sum_{\tau=j}^{t}\|G(\mathbf{x}_\tau)\|_2^2.$$

Since $\|G(\mathbf{x}_\tau)\|_2 \geq \min_{1\leq\tau\leq t}\|G(\mathbf{x}_\tau)\|_2$ and $F(\mathbf{x}_{t+1}) \geq F_*$, then we have

$$\frac{j}{2L}\min_{1\leq\tau\leq t}\|G(\mathbf{x}_\tau)\|_2^2 \leq F(\mathbf{x}_j) - F_*.$$

Hence,

$$\min_{1\leq\tau\leq t}\|G(\mathbf{x}_\tau)\|_2^2 \leq \frac{2L}{j}(F(\mathbf{x}_j) - F_*). \tag{16}$$

We consider three scenarios of $\theta$.

(I). If $\theta > 1/2$, according to Theorem 1, we know that $F(\mathbf{x}_j) - F_*$ converges to 0 in $j = O(c^2 L \epsilon_0^{2\theta-1})$ steps, so $\min_{1\leq\tau\leq t}\|G(\mathbf{x}_\tau)\|_2^2$ converges to 0 in $t = O(c^2 L \epsilon_0^{2\theta-1})$ steps.

(II). If $\theta = 1/2$, let $j = \max(k, 2L)$ and $t = 2j$, where $k = ac^2 L \log\left(\frac{\epsilon_0}{\epsilon^2}\right)$, and $a$ is a constant hided in the big O notation. According to Theorem 1, we have

$$F(\mathbf{x}_k) - F_* \leq \epsilon^2, \tag{17}$$

then the inequality (16), (17) and the choice of $j, k$ yield

$$\min_{1\leq\tau\leq t}\|G(\mathbf{x}_\tau)\|_2^2 \leq \frac{2L}{j}(F(\mathbf{x}_j) - F_*) \leq \epsilon^2,$$

so we know that $t = O(c^2 L \log\left(\frac{\epsilon_0}{\epsilon}\right))$.

(III). If $\theta < 1/2$, let $j$ be an index such that $F(\mathbf{x}_j) - F_* \leq \epsilon'$. We can set $j = 2ac^2 L/\epsilon'^{1-2\theta}$ and thus $t = 4ac^2 L/\epsilon'^{1-2\theta}$, and then we have

$$\min_{1\leq\tau\leq t}\|G(\mathbf{x}_\tau)\|_2^2 \leq \frac{2L}{j}(F(\mathbf{x}_j) - F_*) \leq \frac{\epsilon'\epsilon'^{1-2\theta}}{ac^2} = \frac{\epsilon'^{2-2\theta}}{ac^2}.$$

Let $\epsilon' = c^{\frac{1}{1-\theta}}\epsilon^{\frac{1}{(1-\theta)}}$, we have $\min_{1\leq\tau\leq t}\|G(\mathbf{x}_\tau)\|_2^2 \leq \epsilon^2/a$. We can conclude $t = O(c^{\frac{1}{1-\theta}}L/\epsilon^{\frac{1-2\theta}{1-\theta}})$.

By combining the three scenarios, we can complete the proof. $\qquad\square$

## D   Proof of Lemma 1

*Proof.* The conclusion is trivial when $\mathbf{x} \in \Omega_*$, so we only need to consider the case when $\mathbf{x} \notin \Omega_*$. Define $P_{\eta F}(\mathbf{x}) = \arg\min_{\mathbf{u}} \frac{1}{2}\|\mathbf{u} - \mathbf{x}\|_2^2 + \eta F(\mathbf{u})$.

We first prove for $\theta \in (0, 1/2]$. It is not difficult to see that $\frac{1}{\eta}(\mathbf{x} - P_{\eta F}(\mathbf{x})) \in \partial F(P_{\eta F}(\mathbf{x}))$.

$$D(\mathbf{x}, \Omega_*) \leq \|\mathbf{x} - P_{\eta F}(\mathbf{x})\|_2 + D(P_{\eta F}(\mathbf{x}), \Omega_*)$$

$$\leq \|\mathbf{x} - P_{\eta F}(\mathbf{x})\|_2 + c^{\frac{1}{1-\theta}}\|\partial F(P_{\eta F}(\mathbf{x}))\|_2^{\frac{\theta}{1-\theta}}$$

$$\leq \|\mathbf{x} - P_{\eta F}(\mathbf{x})\|_2 + \frac{c^{\frac{1}{1-\theta}}}{\eta^{\frac{\theta}{1-\theta}}}\|\mathbf{x} - P_{\eta F}(\mathbf{x})\|_2^{\frac{\theta}{1-\theta}}$$

$$\leq \eta(1+L\eta)\|G_\eta(\mathbf{x})\|_2 + c^{\frac{1}{1-\theta}}(1+\eta L)^{\frac{\theta}{1-\theta}}\|G_\eta(\mathbf{x})\|_2^{\frac{\theta}{1-\theta}},$$

where the second inequality uses the result in Lemma 2 and the last inequality follows Proposition 11, which asserts that $\|\mathbf{x} - P_{\eta F}(\mathbf{x})\|_2 \le \eta(1 + L\eta)\|G_\eta(\mathbf{x})\|_2$. Plugging the value $\eta = 1/L$, we have the result.

Next, we prove for $\theta \in (1/2, 1]$. For any $\mathbf{x} \in S_\xi$, we have $P_{\eta F}(\mathbf{x}) \in S_\xi$ and

$$
\begin{aligned}
D(P_{\eta F}(\mathbf{x}), \Omega_*) &\le c(F(P_{\eta F}(\mathbf{x})) - F_*)^\theta \\
&= c(F(P_{\eta F}(\mathbf{x})) - F_*)^{1-\theta}(F(P_{\eta F}(\mathbf{x})) - F_*)^{2\theta-1} \\
&\le c^2\|\partial F(P_{\eta F}(\mathbf{x}))\|_2(F(\mathbf{x}) - F_*)^{2\theta-1} \\
&\le c^2\|\partial F(P_{\eta F}(\mathbf{x}))\|_2\xi^{2\theta-1} \\
&\le c^2(1 + L\eta)\|G_\eta(\mathbf{x})\|_2\xi^{2\theta-1} \\
&\le 2c^2\xi^{2\theta-1}\|G_\eta(\mathbf{x})\|_2,
\end{aligned}
$$

where the second inequality holds because the inequality (13) holds for any $\theta \in (0, 1]$ (by Lemma 2), $F(P_{\eta F}(\mathbf{x})) \le F(\mathbf{x}) \le \xi$, the fourth inequality holds since $\|G_\eta(\mathbf{x})\|_2 \ge \frac{1}{1+L\eta}\|(\mathbf{x} - P_{\eta F}(\mathbf{x}))/\eta\|_2 \ge \frac{1}{1+L\eta}\|\partial F(P_{\eta F}(\mathbf{x}))\|_2$ (by Proposition 11), and the last inequality holds by taking $\eta = 1/L$.

So for $\theta \in (1/2, 1]$ and $\eta = 1/L$, we have

$$
\begin{aligned}
D(\mathbf{x}, \Omega_*) &\le \|\mathbf{x} - P_{\eta F}(\mathbf{x})\|_2 + D(P_{\eta F}(\mathbf{x}), \Omega_*) \\
&\le (\frac{2}{L} + 2c^2\xi^{2\theta-1})\|G(\mathbf{x})\|_2.
\end{aligned}
$$

$\square$

## E Proof of Theorem 5

*Proof.* Let $\mathbf{x}_\delta^*$ be the optimal solution to $\min_{\mathbf{x} \in \mathbb{R}^d} F_\delta(\mathbf{x})$ and $\mathbf{x}_*$ denote an optimal solution to $\min_{\mathbf{x} \in \mathbb{R}^d} F(\mathbf{x})$. Thanks to the strong convexity of $F_\delta(\mathbf{x})$, we have $F_\delta(\mathbf{x}_*) - F_\delta(\mathbf{x}_\delta^*) \ge \frac{\delta}{2}\|\mathbf{x}_* - \mathbf{x}_\delta^*\|_2^2$. Then

$$
F(\mathbf{x}_*) - F(\mathbf{x}_\delta^*) + \delta/2\|\mathbf{x}_* - \mathbf{x}_0\|_2^2 - \delta/2\|\mathbf{x}_\delta^* - \mathbf{x}_0\|_2^2 \ge \delta/2\|\mathbf{x}_* - \mathbf{x}_\delta^*\|_2^2.
$$

Since $F(\mathbf{x}_*) - F(\mathbf{x}_\delta^*) \le 0$, it implies $\|\mathbf{x}_\delta^* - \mathbf{x}_0\|_2 \le \|\mathbf{x}_* - \mathbf{x}_0\|_2$. By Corollary 2, we have

$$
\frac{\eta}{2}\|G_\eta^\delta(\mathbf{x}_{t+1})\|_2^2 \le F_\delta(\mathbf{x}_{t+1}) - F_\delta(\mathbf{x}_\delta^*) \le \frac{L}{2}\|\mathbf{x}_0 - \mathbf{x}_\delta^*\|_2^2\left[1 + \sqrt{\delta/(2L)}\right]^{-2t},
$$

where $\eta \le 1/(L + \delta)$ and $G_\eta^\delta$ is a proximal gradient of $F_\delta(\mathbf{x})$ defined as $G_\eta^\delta(\mathbf{x}) = \frac{1}{\eta}\left(\mathbf{x} - \mathbf{x}_\eta^+(\delta)\right)$ and

$$
\mathbf{x}_\eta^+(\delta) = \arg\min_{\mathbf{y}}\left\{\eta(\nabla f(\mathbf{x}) + \delta(\mathbf{x} - \mathbf{x}_0))^\top(\mathbf{y} - \mathbf{x}) + \eta g(\mathbf{y}) + \frac{1}{2}\|\mathbf{y} - \mathbf{x}\|_2^2\right\}.
$$

Recall that $\mathbf{x}_\eta^+ = P_{\eta g}(\mathbf{x} - \eta\nabla f(\mathbf{x}))$. It is not difficult to derive that $\|\mathbf{x}_\eta^+ - \mathbf{x}_\eta^+(\delta)\|_2 \le 2\eta\delta\|\mathbf{x} - \mathbf{x}_0\|_2$ (by Lemma 4). Since $G_\eta(\mathbf{x}) = \frac{1}{\eta}(\mathbf{x} - \mathbf{x}_\eta^+)$, we have

$$
\|G_\eta(\mathbf{x})\|_2 \le \|G_\eta^\delta(\mathbf{x})\|_2 + \|\mathbf{x}_\eta^+ - \mathbf{x}_\eta^+(\delta)\|_2/\eta \le \|G_\eta^\delta(\mathbf{x})\|_2 + 2\delta\|\mathbf{x} - \mathbf{x}_0\|_2.
$$

Let $\eta = 1/(L + \delta)$, we have

$$
\begin{aligned}
\|G_\eta(\mathbf{x}_{t+1})\|_2 &\le 2\delta\|\mathbf{x}_{t+1} - \mathbf{x}_0\|_2 + \sqrt{L/\eta}\|\mathbf{x}_0 - \mathbf{x}_\delta^*\|_2\left[1 + \sqrt{\delta/(2L)}\right]^{-t} \\
&\le 2\sqrt{2}\delta\|\mathbf{x}_* - \mathbf{x}_0\|_2 + \sqrt{L(L + \delta)}\|\mathbf{x}_0 - \mathbf{x}_*\|_2\left[1 + \sqrt{\delta/(2L)}\right]^{-t}.
\end{aligned}
$$

where we use the inequality $\|\mathbf{x}_\delta^* - \mathbf{x}_0\|_2 \le \|\mathbf{x}_* - \mathbf{x}_0\|_2$. Since $\|G_\eta(\mathbf{x})\|_2$ is a monotonically decreasing function of $\eta$ [7], then $\|G(\mathbf{x})\|_2 \le \|G_\eta(\mathbf{x})\|_2$ for $\eta = 1/(L + \delta) \le 1/L$. Then

$$
\|G(\mathbf{x}_{t+1})\|_2 \le \sqrt{L(L + \delta)}\|\mathbf{x}_0 - \mathbf{x}_*\|_2\left[1 + \sqrt{\delta/(2L)}\right]^{-t} + 2\sqrt{2}\delta\|\mathbf{x}_0 - \mathbf{x}_*\|_2.
$$

$\square$

## F    Proof of Theorem 6

*Proof.*         • We first prove the case when $\theta \in (0, 1/2]$. We can easily induce that $F(\mathbf{x}_k) - F_* \leq$ $\epsilon_0$ from Lemma 3. Let $t_k = \lceil \sqrt{\frac{2L}{\delta_k}} \log \frac{\sqrt{L(L+\delta_k)}}{\delta_k} \rceil$. Applying Theorem 5 to the $k$-the stage of adaAGC and using Lemma 1, we have

$$\|G(\mathbf{x}_{t_k+1}^k)\|_2 \leq (\sqrt{L(L+\delta_k)} \left[ 1 + \sqrt{\frac{\delta_k}{2L}} \right]^{-t_k} + 2\sqrt{2}\delta_k) \tag{18}$$
$$\times (\frac{2}{L}\|G(\mathbf{x}_{k-1})\|_2 + c^{\frac{1}{(1-\theta)}} 2^{\frac{\theta}{(1-\theta)}} \|G(\mathbf{x}_{k-1})\|_2^{\frac{\theta}{(1-\theta)}}),$$

Note that at each stage, we check two conditions (i) $\|G(\mathbf{x}_{\tau+1}^k)\|_2 \leq \varepsilon_{k-1}/2$ and (ii) $\tau = t_k$. If the first condition satisfies first, we proceed to the next stage ($k$ increases by 1). If the second condition satisfies first, then we can claim that $c_e \leq c$ and then we increase $c_e$ by a factor $\gamma > 1$ and then restart the same stage. To verify the claim, assume $c_e > c$ and the second condition satisfies first, i.e., $\tau = t_k$ but $\|G(\mathbf{x}_{\tau+1}^k)\|_2 > \varepsilon_{k-1}/2$. We will deduce a contradiction. To this end, we use (18) and note the value of $t_k$, we have

$$\|G(\mathbf{x}_{t_k+1}^k)\|_2 \leq \left( \delta_k + 2\sqrt{2}\delta_k \right) \times (\frac{2}{L}\|G(\mathbf{x}_{k-1})\|_2 + c^{\frac{1}{(1-\theta)}} 2^{\frac{\theta}{(1-\theta)}} \|G(\mathbf{x}_{k-1})\|_2^{\frac{\theta}{(1-\theta)}})$$
$$\leq 4\delta_k(\frac{2}{L}\|G(\mathbf{x}_{k-1})\|_2 + c^{\frac{1}{(1-\theta)}} 2^{\frac{\theta}{(1-\theta)}} \|G(\mathbf{x}_{k-1})\|_2^{\frac{\theta}{(1-\theta)}})$$
$$\leq \frac{\epsilon_{k-1}}{4} + \frac{c^{\frac{1}{(1-\theta)}} 2^{\frac{\theta}{(1-\theta)}} \epsilon_{k-1}}{4c_e^{\frac{1}{(1-\theta)}} 2^{\frac{\theta}{(1-\theta)}}} \leq \varepsilon_{k-1}/2 = \varepsilon_k,$$

where the last inequality follows that $c_e > c$. This contradicts to the assumption that $\|G(\mathbf{x}_{\tau+1}^k)\|_2 > \varepsilon_{k-1}/2$, which verifies our claim.

Since $c_e$ is increased by a factor $\gamma > 1$ whenever condition (ii) holds first, so within at most $\lceil \log_\gamma(c/c_0) \rceil$ times condition (ii) holds first. Similarly with at most $\lceil \log_2 \varepsilon_0/\epsilon \rceil$ times that condition (i) holds first before the algorithm terminates. We let $T_k$ denote the total number of iterations in order to make condition (i) satisfies in stage $k$. First, we can see that $c_e \leq \gamma c$. Let $\delta_k' = \min(\frac{L}{32}, \frac{\varepsilon_{k-1}^p}{16(\gamma c2^\theta)^{1/(1-\theta)}}) \leq \delta_k$ and $t_k' = \lceil \sqrt{\frac{2L}{\delta_k'}} \log \frac{\sqrt{L(L+\delta_k')}}{\delta_k'} \rceil$. Let $s_k$ denote the number of cycles in each stage in order to have $\|G(\mathbf{x}_{\tau+1}^k)\|_2 \leq \varepsilon_k$. Then $s_k \leq \log_\gamma(c/c_0) + 1$. The total number of iterations of across all stages is bounded by $\sum_{k=1}^K s_k t_k$, which is bounded by

$$\sum_{k=1}^K s_k t_k \leq (1 + \log_\gamma(c/c_0)) \sum_{k=1}^K t_k'.$$

Plugging the value of $t_k'$, we can deduce the iteration complexity in Theorem 6 for $\theta \in (0, 1/2]$.

• Now we consider the proof when $\theta \in (1/2, 1]$. Similar to the proof for $\theta \in (0, 1/2]$, we can easily induce that $F(\mathbf{x}_k) - F_* \leq \epsilon_0$ from Lemma 3. Let $t_k = \lceil \sqrt{\frac{2L}{\delta_k}} \log \frac{\sqrt{L(L+\delta_k)}}{\delta_k} \rceil$. Applying Theorem 5 to the $k$-the stage of adaAGC and using Lemma 1, we have

$$\|G(\mathbf{x}_{t_k+1}^k)\|_2 \leq (\sqrt{L(L+\delta_k)} \left[ 1 + \sqrt{\frac{\delta_k}{2L}} \right]^{-t_k} + 2\sqrt{2}\delta_k) \times (\frac{2}{L} + 2c^2\xi^{2\theta-1})\|G(\mathbf{x}_{k-1})\|_2. \tag{19}$$

Note that at each stage, we check two conditions (i) $\|G(\mathbf{x}_{\tau+1}^k)\|_2 \leq \varepsilon_{k-1}/2$ and (ii) $\tau = t_k$. If the first condition satisfies first, we proceed to the next stage ($k$ increases by 1). If the second condition satisfies first, then we can claim that $c_e \leq c$ and then we increase $c_e$ by a factor $\gamma > 1$ and then restart the same stage. To verify the claim, assume $c_e > c$ and the second condition satisfies first, i.e., $\tau = t_k$ but $\|G(\mathbf{x}_{\tau+1}^k)\|_2 > \varepsilon_{k-1}/2$. We will deduce a contradiction. To this end, we use (19) and note the value of $t_k$, we have

$$\|G(\mathbf{x}_{t_k+1}^k)\|_2 \leq 4\delta_k(\frac{2}{L} + 2c^2\xi^{2\theta-1})\|G(\mathbf{x}_{k-1})\|_2 \leq \frac{\epsilon_{k-1}}{4} + \frac{8c^2\xi^{2\theta-1}}{32c_e^2\epsilon_0^{2\theta-1}}\epsilon_{k-1} \leq \frac{\epsilon_{k-1}}{2} = \epsilon_k,$$

where the last inequality follows that $c_e > c$ and $\xi \le \epsilon_0$. This contradicts to the assumption that $\|G(\mathbf{x}_{\tau+1}^k)\|_2 > \varepsilon_{k-1}/2$, which verifies our claim.

Since $c_e$ is increased by a factor $\gamma > 1$ whenever condition (ii) holds first, so within at most $\lceil \log_\gamma(c/c_0) \rceil$ times condition (ii) holds first. Similarly with at most $\lceil \log_2 \varepsilon_0/\epsilon \rceil$ times that condition (i) holds first before the algorithm terminates. We let $T_k$ denote the total number of iterations in order to make condition (i) satisfies in stage $k$. First, we can see that $c_e \le \gamma c$. Let $\delta_k' = \min(\frac{L}{32}, \frac{1}{32(\gamma c)^2 \epsilon_0^{2\theta-1}}) \le \delta_k$ and $t_k' = \lceil \sqrt{\frac{2L}{\delta_k'}} \log \frac{\sqrt{L(L+\delta_k')}}{\delta_k'} \rceil$. Let $s_k$ denote the number of cycles in each stage in order to have $\|G(\mathbf{x}_{\tau+1}^k)\|_2 \le \varepsilon_k$. Then $s_k \le \log_\gamma(c/c_0) + 1$. The total number of iterations of across all stages is bounded by $\sum_{k=1}^K s_k t_k$, which is bounded by

$$\sum_{k=1}^K s_k t_k \le (1 + \log_\gamma(c/c_0)) \sum_{k=1}^K t_k'.$$

Plugging the value of $t_k'$, we can deduce the iteration complexity in Theorem 6 for $\theta \in (1/2, 1]$.

$\square$

## G   Proof of Theorem 8

First, it is easy to see that in either case, the HEB condition of $F(\cdot)$ with $\theta = 1/2$ and $\mu = \sqrt{2/\mu}$ holds. Next, we prove the following lemma.

**Lemma 5.** *Suppose either $f(\mathbf{x})$ or $g(\mathbf{x})$ satisfies the following property: for any $\mathbf{x} \in dom(F)$, there exists $\mu > 0$ such that*

$$h(\mathbf{x}_*) \ge h(\mathbf{x}) + \partial h(\mathbf{x})^\top (\mathbf{x}_* - \mathbf{x}) + \frac{\mu}{2}\|\mathbf{x} - \mathbf{x}_*\|_2^2, \tag{20}$$

*where $\mathbf{x}_*$ is the closest optimal solution to $\mathbf{x}$. Then we have the following:*

$$F(\mathbf{x}_+) - F(\mathbf{x}_*) \le O(1/\mu)\|G(\mathbf{x})\|_2^2$$

*where*

$$\mathbf{x}_+ = \arg\min_{\mathbf{u} \in \mathbb{R}^d} \left[ f(\mathbf{x}) + \langle \nabla f(\mathbf{x}), \mathbf{u} - \mathbf{x} \rangle + \frac{L}{2}\|\mathbf{u} - \mathbf{x}\|_2^2 + g(\mathbf{u}) \right],$$
$$G(\mathbf{x}) = L(\mathbf{x} - \mathbf{x}_+),$$

*Proof.* Define $\phi(\mathbf{u}) = f(\mathbf{x}) + \langle \nabla f(\mathbf{x}), \mathbf{u} - \mathbf{x} \rangle + \frac{L}{2}\|\mathbf{u} - \mathbf{x}\|_2^2 + g(\mathbf{u})$ and then $\partial \phi(\mathbf{u}) = \nabla f(\mathbf{x}) + L(\mathbf{u} - \mathbf{x}) + \partial g(\mathbf{u})$. By the first-order optimality condition of $\mathbf{x}_+$, for all $\mathbf{u} \in dom(F)$ there exists $\mathbf{v}_+ \in \partial g(\mathbf{x}_+)$:

$$\langle \nabla f(\mathbf{x}) + \mathbf{v}_+ - G(\mathbf{x}), \mathbf{u} - \mathbf{x}_+ \rangle \ge 0,$$

Without loss of generality, we first assume $f(\cdot)$ and $g(\cdot)$ both satisfy (20) with $\mu_f \ge 0$ and $\mu_g \ge 0$. When $\mu_f = 0$ or $\mu_g = 0$, the inequality is automatically satisfied. Then we have

$$f(\mathbf{x}_*) - \frac{\mu_f}{2}\|\mathbf{x}_* - \mathbf{x}\|_2^2 \ge f(\mathbf{x}) + \langle \nabla f(\mathbf{x}), \mathbf{x}_* - \mathbf{x} \rangle$$
$$= f(\mathbf{x}) + \langle \nabla f(\mathbf{x}), \mathbf{x}_+ - \mathbf{x} \rangle + \langle \nabla f(\mathbf{x}), \mathbf{x}_* - \mathbf{x}_+ \rangle$$
$$\ge f(\mathbf{x}) + \langle \nabla f(\mathbf{x}), \mathbf{x}_+ - \mathbf{x} \rangle + \langle G(\mathbf{x}), \mathbf{x}_* - \mathbf{x}_+ \rangle + \langle \mathbf{v}_+, \mathbf{x}_+ - \mathbf{x}_* \rangle$$
$$\ge f(\mathbf{x}) + \langle \nabla f(\mathbf{x}), \mathbf{x}_+ - \mathbf{x} \rangle + \langle G(\mathbf{x}), \mathbf{x}_* - \mathbf{x}_+ \rangle + g(\mathbf{x}_+) - g(\mathbf{x}_*) + \frac{\mu_g}{2}\|\mathbf{x}_+ - \mathbf{x}_*\|^2$$
$$= \phi(\mathbf{x}_+) - \frac{L}{2}\|\mathbf{x} - \mathbf{x}_+\|_2^2 + \langle G(\mathbf{x}), \mathbf{x}_* - \mathbf{x}_+ \rangle + \frac{\mu_g}{2}\|\mathbf{x}_+ - \mathbf{x}_*\|^2 - g(\mathbf{x}_*)$$
$$= \phi(\mathbf{x}_+) - \frac{1}{2L}\|G(\mathbf{x})\|_2^2 + \langle G(\mathbf{x}), \mathbf{x}_* - \mathbf{x}_+ \rangle + \frac{\mu_g}{2}\|\mathbf{x}_+ - \mathbf{x}_*\|^2 - g(\mathbf{x}_*),$$

where the second inequality uses the optimality condition of $\mathbf{x}_+$ and the third inequality uses the condition (20) of $g(\cdot)$. Next, we consider two cases.

Case I: $\mu_g > 0$ and $\mu_f \geq 0$ (i.e., $g(\cdot)$ satisfies (20)). We have

$$f(\mathbf{x}_*) \geq \phi(\mathbf{x}_+) - \frac{1}{2L}\|G(\mathbf{x})\|_2^2 - \frac{1}{2\mu_g}\|G(\mathbf{x})\|_2^2 - \frac{\mu_g}{2}\|\mathbf{x}_* - \mathbf{x}_+\|_2^2 + \frac{\mu_g}{2}\|\mathbf{x}_+ - \mathbf{x}_*\|^2 - g(\mathbf{x}_*)$$

As a result,

$$F(\mathbf{x}_*) \geq \phi(\mathbf{x}_+) - \frac{1}{2L}\|G(\mathbf{x})\|_2^2 - \frac{1}{2\mu_g}\|G(\mathbf{x})\|_2^2 \geq F(\mathbf{x}_+) - \frac{1}{2L}\|G(\mathbf{x})\|_2^2 - \frac{1}{2\mu_g}\|G(\mathbf{x})\|_2^2$$

Thus

$$F(\mathbf{x}_+) - F(\mathbf{x}_*) \leq \frac{L + \mu_g}{2L\mu_g}\|G(\mathbf{x})\|_2^2$$

Case II: $\mu_f > 0$ and $\mu_g \geq 0$ (i.e., $f(\cdot)$ satisfies (20)). Then we have

$$f(\mathbf{x}_*) \geq \phi(\mathbf{x}_+) - \frac{1}{2L}\|G(\mathbf{x})\|_2^2 + \langle G(\mathbf{x}), \mathbf{x}_* - \mathbf{x}_+ \rangle + \frac{\mu_f}{2}\|\mathbf{x}_* - \mathbf{x}\|_2^2 - g(\mathbf{x}_*)$$

$$\geq \phi(\mathbf{x}_+) - \frac{1}{2L}\|G(\mathbf{x})\|_2^2 + \langle G(\mathbf{x}), \mathbf{x}_* - \mathbf{x} \rangle + \langle G(\mathbf{x}), \mathbf{x} - \mathbf{x}_+ \rangle + \frac{\mu_f}{2}\|\mathbf{x}_* - \mathbf{x}\|_2^2 - g(\mathbf{x}_*)$$

$$\geq \phi(\mathbf{x}_+) + \frac{1}{2L}\|G(\mathbf{x})\|_2^2 + \langle G(\mathbf{x}), \mathbf{x}_* - \mathbf{x} \rangle + \frac{\mu_f}{2}\|\mathbf{x}_* - \mathbf{x}\|_2^2 - g(\mathbf{x}_*)$$

$$\geq \phi(\mathbf{x}_+) + \frac{1}{2L}\|G(\mathbf{x})\|_2^2 - \frac{1}{2\mu_f}\|G(\mathbf{x})\|_2^2 - \frac{\mu_f}{2}\|\mathbf{x}_* - \mathbf{x}\|_2^2 + \frac{\mu_f}{2}\|\mathbf{x}_* - \mathbf{x}\|_2^2 - g(\mathbf{x}_*)$$

$$\geq F(\mathbf{x}_+) - \frac{1}{2\mu_f}\|G(\mathbf{x})\|_2^2 - g(\mathbf{x}_*)$$

Thus,

$$F(\mathbf{x}_+) - F(\mathbf{x}_*) \leq \frac{1}{2\mu_f}\|G(\mathbf{x})\|_2^2$$

In either case, we have $F(\mathbf{x}_+) - F(\mathbf{x}_*) \leq O(1/\mu)\|G(\mathbf{x})\|_2^2$. $\qquad\square$

Finally, we see that in order to guarantee $F(\mathbf{x}_+) - F(\mathbf{x}_*) \leq \epsilon$, we need to have $\|G(\mathbf{x})\|_2 \leq O(\sqrt{\mu\epsilon})$.