[Reviews · NeurIPS 2017]

Reviewer 1



The paper extends the linear convergence result for PG and rAPG from QGC to a more broader condition termed here as HEB. An adaptive version of APG is also presented that does not insist on knowing the "condition number" apriori. The extension to HEB and the convergence results presented are all interesting (did not fully check correctness of the proofs). As an example of ML application a case where \theta=0.5 was shown (Cor 1). However, since it is \theta strictly less than 0.5 that is more interesting, are there examples of ML set-ups where this is the case? I have read the rebuttal and would like to stay with the score.

Reviewer 2



Summary: This paper studies the convergence rate of the proximal gradient and its accelerated version under the more general Holderian error bound condition, further extending a line of recent efforts that has weakened the usual strong convexity assumption to the quadratic growth condition. The main contribution is a restarting scheme that eliminates the need of knowing the Holderian constant. The authors complement their theoretical claim with some numerical experiments, which seem to verify the efficiency of the proposed adaptive algorithm. As far as I can tell, this work is a fine synthesis of some recent developments around semialgebraic functions, error bound, and the proximal gradient algorithm. The results are not too surprising given the existing known results. Nevertheless, it is a very welcome addition to the field. And given its practical nature, the proposed algorithm might be useful for a variety of applications (some toy demonstrations in the experiment). Some comments: While the technical proofs involve many pieces from relevant references, the general idea is in fact quite simple (and dates back at least to FISTA): one first derives some condition (e.g. Theorem 5) that is guaranteed to hold if the correct parameter is used, then in the algorithm one periodically checks this condition; if it ever falls short then we know our parameter of choice is too small and we increase it (the familiar doubling trick) and rerun the algorithm. This high level description of the adaptive algorithm perhaps worth explicitly mentioning in the revision (to give the less familiar readers a clear guidance). Line 85: no need to mention that "g(x) > -\infty for all x" since the range is assumed to be (-\infty, \infty]. Theorem 1: the option I and II are only defined in the appendix. better mention them in the main text (or at least say "see appendix"). The calculations and notations in the proofs of Theorem 1 and 2 are a little sloppy: when theta > 1/2, the rate does not depend on epsilon (the target accuracy) at all? Some more details between say line 68 and line 69 are needed. Section 4: Can one develop a similar proof for the objective value gap? Is it because of the DUAL algorithm you use to solve Eq (8) that limits the guarantee to the minimal gradient? Can we say anything about the dependence of the constants theta and c on the dimension and the number of samples? It could be possible that a linear rate is in fact worse than a sublinear rate if such dependence is not carefully balanced. Experiments: On some datasets (cpusmall and bodyfat), why proximal gradient takes so many more iterations? This looks a little unusual. How many of the proximal mappings are spent on searching for the step size (for each method)? Also, to give a fuller picture, maybe also include the result for the decrease of the objective values. Another observation here: is it really meaningful to consider an absolute epsilon = 10^{-6} say? For optimization per se, sure. But what if say we only care about the classification accuracy? Would it be possible that an solution at epsilon = 10^{-3} already gives us comparable classification accuracy? (This should be easy to verify since the authors did experiment with a classification dataset.)